# BRCA1-associated structural variations are a consequence of polymerase theta-mediated end-joining

J. A. Kamp [1], R. van Schendel [1], I. W. Dilweg [1] & M. Tijsterman [1,2 ✉]

Failure to preserve the integrity of the genome is a hallmark of cancer. Recent studies have revealed that loss of the capacity to repair DNA breaks via homologous recombination (HR) results in a mutational profile termed BRCAness. The enzymatic activity that repairs HR substrates in BRCA-deficient conditions to produce this profile is currently unknown. We here show that the mutational landscape of BRCA1 deficiency in *C. elegans* closely resembles that of BRCA1-deficient tumours. We identify polymerase theta-mediated end-joining (TMEJ) to be responsible: knocking out *polq-1* suppresses the accumulation of deletions and tandem duplications in *brc-1* and *brd-1* animals. We find no additional back-up repair in HR and TMEJ compromised animals; non-homologous end-joining does not affect BRCAness. The notion that TMEJ acts as an alternative to HR, promoting the genome alteration of HR-deficient cells, supports the idea that polymerase theta is a promising therapeutic target for HR-deficient tumours.

[1] Department of Human Genetics, Leiden University Medical Center, Einthovenweg 20, 2333 ZC Leiden, The Netherlands. [2] Institute of Biology Leiden, Leiden University, Sylviusweg 72, 2333 BE Leiden, The Netherlands. ✉email: m.tijsterman@lumc.nl

The integrity of genetic information, which is of vital importance for life, is dependent on accurate DNA replication and multiple mechanisms that repair damaged DNA. Failure of proper genome maintenance can provoke high numbers of mutations, which can provide cells with a selective growth advantage, eventually leading to cancer[1]. Genome instability is not necessarily only causal to cancer, it also provides an Achilles' heel: because cancer cells frequently have acquired the ability to progress through the cell cycle in the presence of DNA damage they are vulnerable to DNA damaging agents—cell cycle progression with underreplicated genomes produces aneuploidy and cell death. Multiple cancer therapies, such as DNA damage-inflicting irradiation or chemotherapy, are aimed at this vulnerability, however, emerging treatment strategies are aimed to take advantage of disturbed DNA repair systems that may be present in different tumour types. For instance, treatment with PARP inhibitors, which results in increased numbers of double-strand DNA breaks (DSBs), is clinically effective in carcinomas that are deficient in homologous recombination (HR)[2], the major error-free pathway that repairs DSBs by predominantly using the sister chromatid as a undamaged template for repair. Careful assessment of the repair capacity of tumour cells can thus influence treatment choice but can also prevent overtreatment, if upfront analysis would predict treatment unresponsiveness. It is thus becoming increasingly important to determine the genetic make-up of a tumour to reveal potential vulnerabilities or insensitivities.

Recent studies on sequenced tumour material have revealed that the profile of mutations, so-called mutational signatures, can be used as a biomarker for DNA repair deficiencies[3–6]. HR deficiency results in particular mutagenic outcomes, which have been incorporated in a recently developed model, called HRDetect, which is able to predict HR deficiency in tumours using sequenced genomes as input. The used algorithms also are able to distinguish BRCA1- from BRCA2-deficient tumours[7]—while both BRCA proteins are vital to repair DSBs through HR, they have very different roles. The signatures included in this model are base substitutions, tandem duplications, deletions smaller than 100 kb and micro-homology-mediated deletions[7]. To explain the increased base substitution rate that is observed in BRCA1-deficient cells, loss of suppression of translesion synthesis has been proposed[8,9]. Tandem duplications are structural variations that presumably arise from repair of DSBs[10–12] or stalled replication forks[13] via head-to-tail duplication of genetic information. Deletions are losses of genetic information that can result from error-prone repair of DSBs; when repair is guided by annealing of complementary nucleotides present at the break ends, the deletion product will have so-called micro-homology at the junction[14].

Because error-free HR is impaired in BRCA1- and BRCA2-deficient cells[15,16], it is likely that error-prone repair of HR substrates are causal for the mutations observed in these tumours. It is often postulated that in the absence of HR, DNA double-strand breaks are repaired via non-homologous end-joining (NHEJ)[17–20], however, NHEJ is not guided by DNA sequence and does not typically produce deletions featuring micro-homologies. Alternatively, repair of breaks that are aligned to be processed by HR is, in its absence, performed by an alternative end-joining pathway. Previous work has demonstrated that micro-homology-mediated deletions are often a result of polymerase theta action[10,21–24]. Recent work also demonstrated that replication-associated breaks are repaired by polymerase theta-mediated end-joining (TMEJ), instead of HR, in situations where undamaged sister chromatids are not available to serve as a template[25,26]. Polymerase theta is therefore a logical candidate to produce the micro-homology-mediated deletions in BRCA-deficient cells.

To test this hypothesis, we here investigate spontaneous mutagenesis in *C. elegans* defective for *brc-1* (*BRCA1* ortholog). Although mammalian BRCA1-deficient cells can only proliferate when genome integrity is further compromised by altering the p53 status[27], *brc-1*-deficient worms develop normally and are fertile. The *C. elegans* model system thus provides us with a clean genetic context to study BRCA1 deficiency, alone or in combination with deficiencies in other repair factors. We find that *brc-1*-deficient animals accumulate mutations similar to BRCA1-deficient tumours (micro-homology-mediated deletions, tandem duplications and base substitutions), and we causally implicate polymerase theta in the emergence of structural variations. Our data demonstrate that polymerase theta acts as an alternative to HR by repairing HR intermediates, thus protecting the integrity of the genome but with mutations as a consequence.

## Results

**A BRCAness mutational profile in *C. elegans*.** To investigate mechanisms of DNA mutagenesis, we perform mutation accumulation sequencing experiments with DNA repair defective nematodes: after propagating animals for 40–60 generations (Supplementary Table 3), their 100 million base pair (bp)-sized genomes[28] are sequenced. Unique mutations that arise in the germline during prolonged culturing are identified by comparing the genomes of the propagated strains to the genome at the start of the experiment.

To determine the contribution of BRCA1-mediated HR on genome stability and the suppression of mutagenesis, we clonally propagated *brc-1* mutant animals while monitoring the number of generations. In addition to *brc-1* mutant animals, we also propagated null mutants for BRC-1's binding partner BARD1/BRD-1, whose heterodimerisation with BRC-1 is essential for BRC-1 stability[29]. Indeed, homology-directed repair in somatic cells was decreased to the same extent in *brd-1* mutants as in *brc-1* mutants, assessed by a DR-GFP reporter system we previously developed[30], which monitors homology-directed repair of IsceI-induced DSBs in intestinal nuclei[30] (Supplementary Fig. 1). By sequencing the genomes of *brd-1* animals in parallel to *brc-1* animals, we can assess whether BRC-1 and BRD-1 have independent roles in the maintenance of genome stability in the germline.

Strikingly, we found that both mutants accumulate 8–10 fold more deletions and deletions–insertions (deletions with an accompanying insertion) than wild-type nematodes (Fig. 1a, c)[31]. Although wild-type worms on average obtain 1 deletion per 30 generations, *brc-1* and *brd-1* mutants obtain a deletion every 3–4 generations, indicating that at least a subset of HR substrates are now shuttled towards an error-prone repair pathway. We found no significant difference in deletion rate between *brc-1* and *brd-1* ($p = 0.236$), suggesting BRC-1 and BRD-1 do not have independent roles in deletion prevention. The size of the deletions that accumulate in *brc-1* and *brd-1* mutants are within a rather narrow range: 77% are smaller than 30 bp (Fig. 1a). The deletions without an insertion are characterised by an overrepresentation of micro-homology: 79% of deletions had at least one nucleotide that could be mapped to either junction (Fig. 1a; Supplementary Fig. 3), whereas 47% results from an in silico generated random set of *C. elegans* deletions[25]. Furthermore, many deletions also contained inserted nucleotides: 27 out of 90 for *brc-1* and 19 out of 55 for *brd-1*. Of all 46 deletions–insertions identified in the genomes of *brc-1* and *brd-1*-deficient nematodes, 33 of the insertions were at least 5 nucleotides long, allowing inspection of their origin and potential mapping to flanking sequences with sufficient reliability. Of these insertions, 24 (73%) were identical to sequences in the immediate

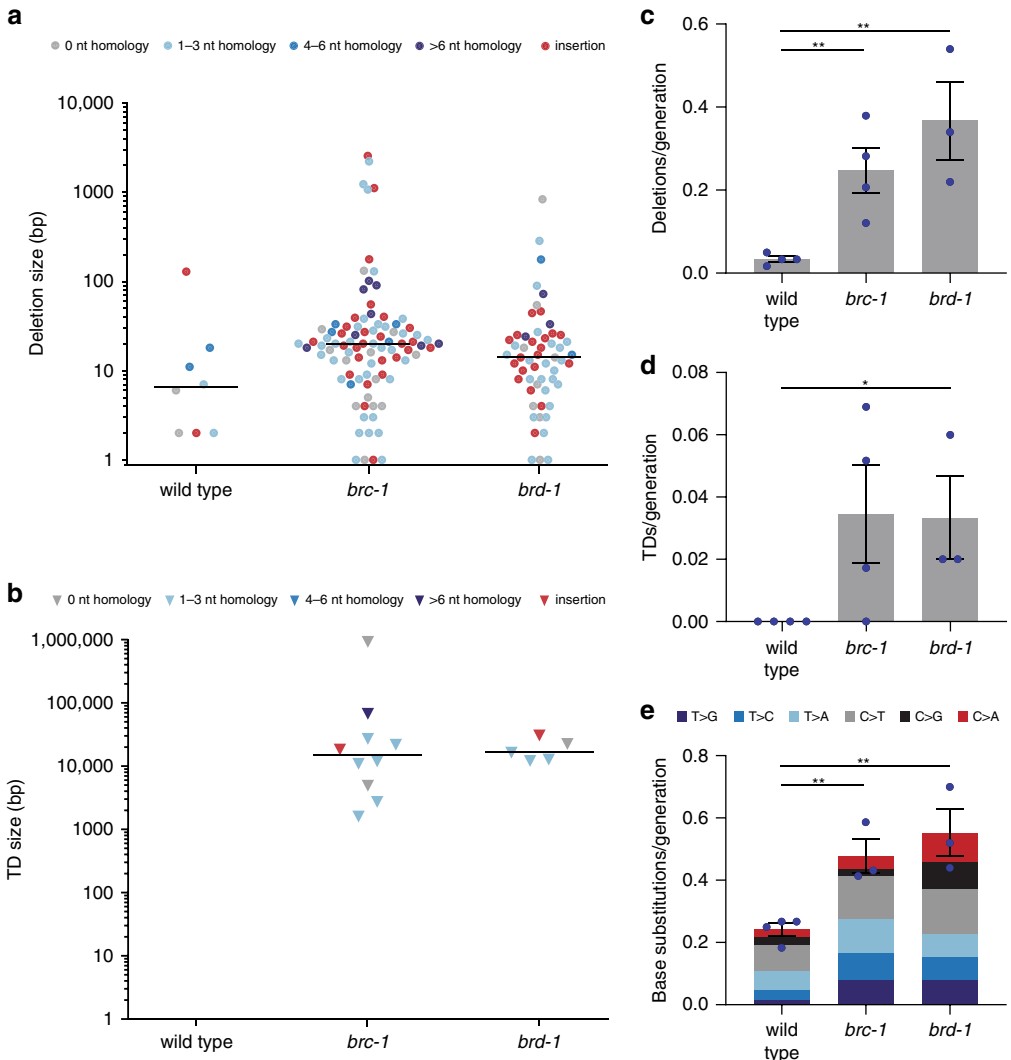

**Fig. 1 brc-1 and brd-1-deficient nematodes accumulate mutations in their genomes. a** Size and junction representation of deletions that were obtained in wild-type ($n = 4$), brc-1 ($n = 6$) and brd-1 ($n = 3$) mutant animals. Deletions without homology are marked in grey, deletions with homology are marked in blue. Increasing homology size is depicted by increased colour intensity. Deletions with insertions are marked in red. The median deletion sizes are indicated by horizontal lines. **b** Size and junction representation of tandem duplication (TD) that were obtained in wild-type, brc-1 and brd-1 mutant animals. TDs without homology are shown in grey, TDs with homology are marked in blue. Increasing homology size is depicted by increased colour intensity. TDs with insertions are marked in red. The median TD sizes are indicated by horizontal lines. **c** Quantification of the average rate of deletions per generation in animals of different genotypes. The rate is defined as the number of deletions divided by the number of propagated generations per animal. The rate per strain is represented in blue dots. Two-tailed t-tests were used to determine statistical significance ($*P < 0.05$, $**P < 0.01$). **d** Quantification of the average rate of TDs per generation in animals of different genotypes. The rate is defined as the number of TDs divided by the number of propagated generations per animal. The rate per strain is represented in blue dots. Two-tailed t-tests were used to determine statistical significance ($*P < 0.05$, $**P < 0.01$). **e** A representation of base substitution type and the average rate of base substitutions per generation in animals of different genotypes. The different types of base substitutions are labelled with different colours. The rate is defined as the number of base substitutions divided by the number of propagated generations per animal. The rate per strain is represented in blue dots. Two-tailed t-tests were used to determine statistical significance ($*P < 0.05$, $**P < 0.01$). Error bars represent SEM. Wild type: $n = 4$ of 60 generations, brc-1: $n = 6$ of 58 generations, brd-1: $n = 3$ of 50 generations.

vicinity of their cognate deletion junction and are thus characterised as templated insertions[32].

Apart from deletions, we found that brc-1 and brd-1 mutant animals also accumulate tandem duplications (TDs). Although we have not observed any TD in 240 generations of wild-type animals (Fig. 1b, d), we found 10 in 300 generations of brc-1 animals and 5 in 150 generations of brd-1 animals (Fig. 1b, d; Supplementary Fig. 5). The sizes of the duplicated segments ranged from 1 kb to 1 Mb, but the majority were ~10–20 kb in size. The rate of TDs in brc-1 and brd-1 mutants is approximately tenfold lower than the rate of deletions in these mutants, implying that either the DNA damage leading to a TD is less

frequent than a deletion-inducing DSB, or that a deletion is a more likely outcome of DSB repair than a TD. The junctional features are nevertheless very similar being characterised by micro-homology and the occasional presence of insertions. This similarity suggests that the same mechanism that is responsible for generating a deletion is involved in (a likely late step of) TD formation.

Besides an increase in structural variations, we also found a small but statistically significant increase in base substitutions in brc-1 and brd-1 mutants as compared to wild-type animals (Fig. 1e), and no significant difference between brc-1 and brd-1 ($p = 0.464$). No apparent sequence motif dominated the spectrum

(Fig. 1e; Supplementary Fig. 4), leaving no hints towards the source of the increased SNV mutagenesis.

Taken together, our observations show that brc-1 and brd-1 mutants accumulate structural variations and base substitutions at higher rates than wild-type nematodes.

**NHEJ does not affect C. elegans BRCAness.** The inheritable deletions and TDs that accumulate in the genomes of brc-1 and brd-1 animals are likely a consequence of DSB repair by end-joining. One pathway previously proposed to be responsible for deletion mutagenesis is NHEJ[33]. Indeed, we have previously shown that NHEJ can act to process meiotic DSBs in the C. elegans germline in animals that are deficient for the CtiP homologue COM-1, a protein that is required for HR by stimulating DNA end-resection[34]. To assess a potential causal role for this pathway in the aetiology of structural variations in brc-1 and brd-1 mutants, null alleles of the genes encoding C. elegans LIG4 (lig-4) and KU80 (cku-80) were crossed into brc-1 and brd-1 strains. Three lines of the acquired double mutant animals, as well as three lines of lig-4 and cku-80 single mutant animals were propagated for 50 generations, after which their genomes were sequenced.

Unexpectedly, we observed an increase in deletion mutagenesis in cku-80 and lig-4 mutant animals as compared to wild-type animals (Fig. 2c). This finding argues that besides HR, also NHEJ factors contribute to error-free repair of DNA damage in the germline of C. elegans, albeit a small contribution compared to the contribution of brc-1 and brd-1. Also, the base substitution rate is increased in cku-80 and lig-4 mutants as compared to wild type (Fig. 2e). However, we found that NHEJ is not the back-up pathway for repairing HR intermediates in brc-1 and brd-1-deficient animals: no significant reduction of the deletion rate is observed when brc-1 lig-4 is compared to brc-1 nor when brd-1 cku-80 is compared with brd-1 (Fig. 2c). In agreement, the deletion size and the degree of micro-homology at the deletion junctions is similar in NHEJ-proficient versus NHEJ-deficient conditions. The same conclusion pertains to TDs: both the rate and size of TDs, as well as the degree of micro-homology at the junction is undistinguishable in NHEJ-deficient and NHEJ-proficient brc-1 and brd-1 animals (Fig. 2b–d).

Together, these data demonstrate that NHEJ is not responsible for nor contributes to the mutagenesis that results from defective HR.

**TMEJ produces the structural variations accumulating in brc-1/brd-1.** The demonstrated lack of NHEJ involvement argues for an alternative end-joining pathway to produce the mutations that accumulate in brc-1 and brd-1 animals. Insertions and micro-homology at deletion junctions are signature features of TMEJ hinting towards a causal involvement of polymerase theta in the formation of brc-1 and brd-1-associated deletions. To test this hypothesis, brc-1 polq-1 and brd-1 polq-1 double mutants were generated. Although these double mutants are viable and fertile, animals have reduced brood size and increased embryonic lethality preluding a compensatory interaction of these genes. We subsequently propagated multiple independent lines of double mutant animals and sequenced three lines that were maintained for 40 generations. A progressive decline in animal fertility made it difficult to maintain healthy cultures for even more generations.

Strikingly, we observed an almost complete loss of deletion formation in brc-1 polq-1 and brd-1 polq-1 animals, as compared with brc-1 and brd-1 (Fig. 3a, c). While brc-1 and brd-1 animals together accumulated 134 deletions that were smaller than 100 bp after 498 generations in total, the genomes of brc-1 polq-1 and brd-1 polq-1 animals together contained only 9 of these small

deletions after 240 generations—a sevenfold decrease. Interestingly, small deletions, while severely reduced, were not entirely absent in double mutant animals, and importantly, those that remained to be induced are characterised by extensive homology at the junction sites (Fig. 3a; Supplementary Fig. 3). Our data demonstrate that all small deletions with ≤6 nt of homology are the result of polymerase theta action on DSBs and argues for the existence of a mechanism that can repair DSBs with >6 nt homology in an polymerase theta-independent manner. We also observed a small number of very large deletions (>10 kb), which is in line with previous data that demonstrated extensive loss of DNA at TMEJ substrates in the absence of polymerase theta[31]. Deletions of that size will affect multiple genes in a gene-dense organism such as C. elegans and their presence is likely counterselected in animal propagation experiments, which also explains the observed loss of population fecundity.

For TDs, the pattern is identical: a dramatic drop in rate in brc-1 polq-1 and brd-1 polq-1 mutants, the only residual case having 13 bp of sequence homology at the junction (Fig. 3b, d). Thus, both types of structural variations, i.e. deletions and TDs, which spontaneously arise in animals that have a defect in HR, are the result of polymerase theta action. In contrast, the increased number of base substitutions in brd-1 and brc-1 mutants is not related to TMEJ activity (Fig. 3e, f).

**TMEJ can repair ionising radiation-induced HR substrates.** Although genetic inactivation of BRCA1 in mammalian cells prohibits cell proliferation, the absence of its ortholog in nematodes is tolerated. One could suggest that TMEJ is more active in C. elegans than in mammalian cells, and thereby fully compensates for loss of HR. We consider this explanation unlikely for two reasons: (i) mammalian cells that tolerate BRCA1 deficiency because of additional tumour promoting mutants are reliant for their survival on TMEJ; under these conditions TMEJ activity is thus sufficient, (ii) C. elegans brc-1 mutants survive TMEJ impairment. Another explanation for why BRCA1 knockouts worms are viable, whereas mammalian knockout cells are not, may be a lack of HR substrates in worms grown under laboratory conditions.

To further investigate a potential synthetic interaction of HR and TMEJ we exposed brc-1 polq-1 and brc-1 polq-1 animals to ionising radiation (IR) as IR leads to DSBs[35,36]. As a proxy for DNA repair, we quantified the survival of the irradiated nematodes' progeny (Fig. 4). As previously described, inactivation of TMEJ only mildly affects IR sensitivity[37], whereas NHEJ inactivation has no influence whatsoever, pointing to a more prominent role of HR to remove DSBs in germ cells (the target tissue in this assay)[38]. Indeed, in line with previous work, brc-1 inactivation reveals a marked hypersensitivity to IR[39]. From the synergistic increase in hypersensitivity of brc-1 polq-1 double mutants to IR, we conclude that TMEJ can compensate for loss of HR activity. Confirming the lack of interaction between NHEJ and HR, we found that brc-1 lig-4 double mutant animals were as sensitive to IR as brc-1 single mutant animals. These data support the mutagenesis data in the sense that TMEJ, and not NHEJ, is responsible for the residual repair of DSBs in a BRC-1-deficient context.

**Discussion**

Here, we show in a genetic system tolerating BRCA1 deficiency that HR impairment results in three different classes of mutations that are also observed in HR-deficient tumour cells: (i) small deletions with overrepresentation of micro-homology at the junction, (ii) duplications of ~10 kb that are located immediately adjacent to their counterpart, (iii) single nucleotide variants. We

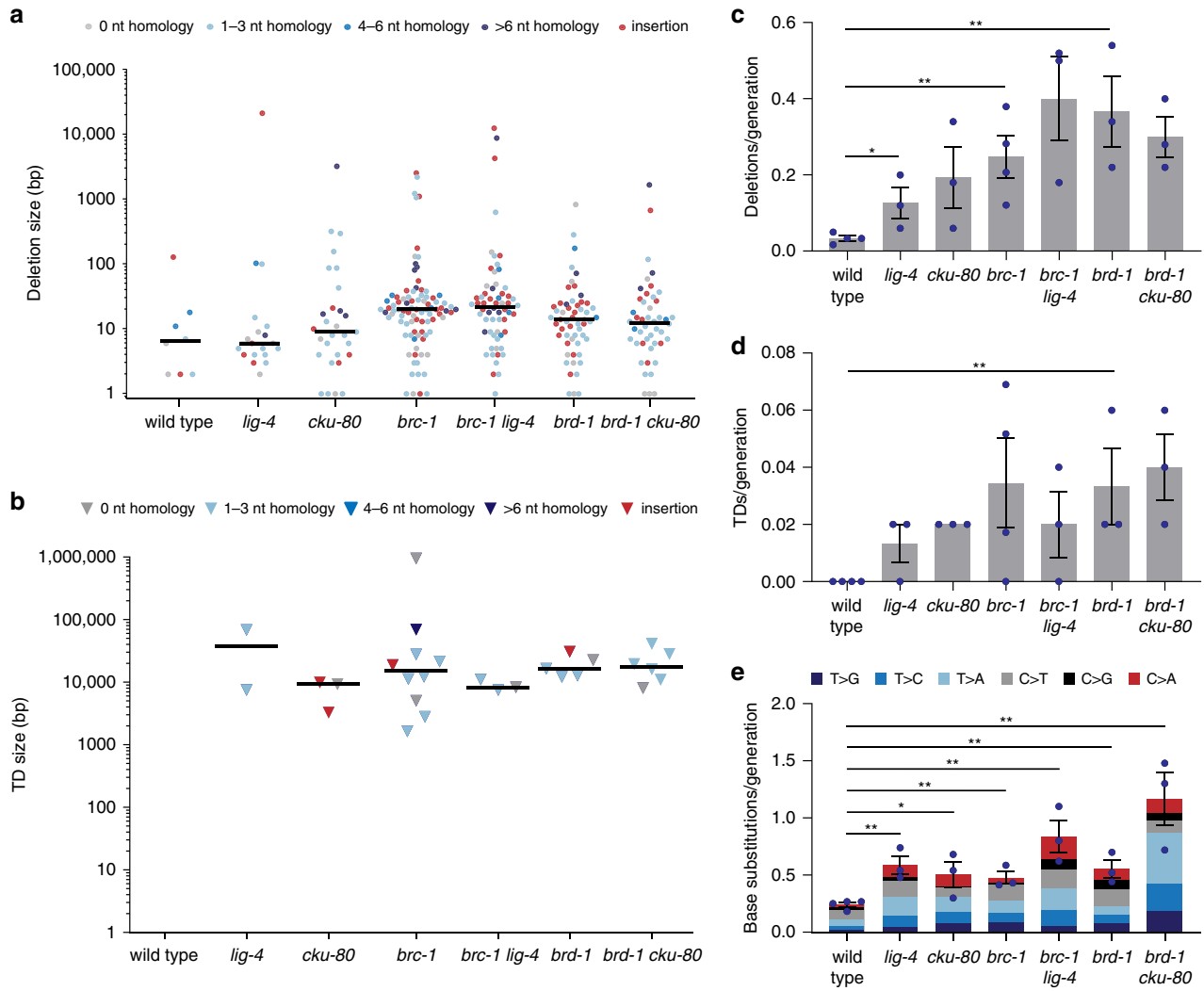

**Fig. 2 NHEJ is not responsible for mutations in the genomes of brc-1 and brd-1 animals. a** Size and junction representation of deletions that were obtained in wild-type and mutant animals ($n = 3$ for all NHEJ-deficient strains). Deletions without homology are shown in grey, deletions with homology are marked in blue. Increasing homology size is depicted in increasing colour intensity. Deletions with insertions are marked in red. The median deletion sizes are indicated by horizontal lines. **b** Size and junction representation of TDs that were obtained in wild-type and mutant animals. TDs with homology at the junctions are marked in blue. Increasing homology size is depicted in increasing colour intensity. TDs with insertions are marked in red. The median TD sizes are indicated by horizontal lines. **c** Quantification of the average rate of deletions per generation in animals of different genotypes. The rate is defined as the number of deletions divided by the number of propagated generations per animal. The rate per strain is represented in blue dots. Two-tailed $t$-tests were used to determine statistical significance (*$P < 0.05$, **$P < 0.01$). **d** A quantification of the average rate of TDs per generation in animals of different genotypes. The rate is defined as the number of TDs divided by the number of propagated generations per animal. The rate per strain is represented in blue dots. Two-tailed $t$-tests were used to determine statistical significance (*$P < 0.05$, **$P < 0.01$). **e** A representation of base substitution type and the average rate of base substitutions per generation in animals of different genotypes. The different types of base substitutions are labelled with different colours. The rate is defined as the number of base substitutions divided by the number of propagated generations per animal. The rate per strain is represented in blue dots. Two-tailed $t$-tests were used to determine statistical significance (*$P < 0.05$, **$P < 0.01$). Error bars represent SEM. Wild type: $n = 4$ of 60 generations, brc-1: $n = 6$ of 58 generations, cku-80, lig-4, brd-1, brc-1 lig-4 and brd-1 cku-80: $n = 3$ of 50 generations.

subsequently demonstrate that the structural variations are the result of TMEJ, and not NHEJ, acting on HR substrates.

The molecular configuration of TMEJ products in HR compromised conditions provide valuable clues, as well as restrictions, as to the role of BRCA1/BARD1 in HR. A priori, a number of discrete steps can be described for HR, for which different DNA or DNA-protein intermediates can be envisioned onto which alternative repair could principally act: (i) end-resection of break ends followed by coating of the DNA with the ssDNA binding protein RPA, (ii) replacement of RPA by the recombinase RAD51, a process stimulated by BRCA2, (iii) invasion of the homologous template/sister chromatid followed by D-loop

extension, (iv) resolution of the extended break end via different mechanisms, one of which being SDSA, in which the extended DSB end is suggested to anneal to the end-resected other end of the break. Multiple roles for mammalian BRCA1 have been put forward, particularly in step (i): in the absence of BRCA1, CtIP-mediated resection of DSB ends is reduced[40,41], and step (iii): RAD51-mediated strand invasion is enhanced by the BRCA1/BARD-1 heterodimer[42]. Our data is consistent with either one of these roles, but importantly, the fact that TDs appear in brc-1/brd-1 animals argues that strand invasion and D-loop extension is not completely impaired—we favour the interpretation that TDs are the product of joining one DSB end that is first extended

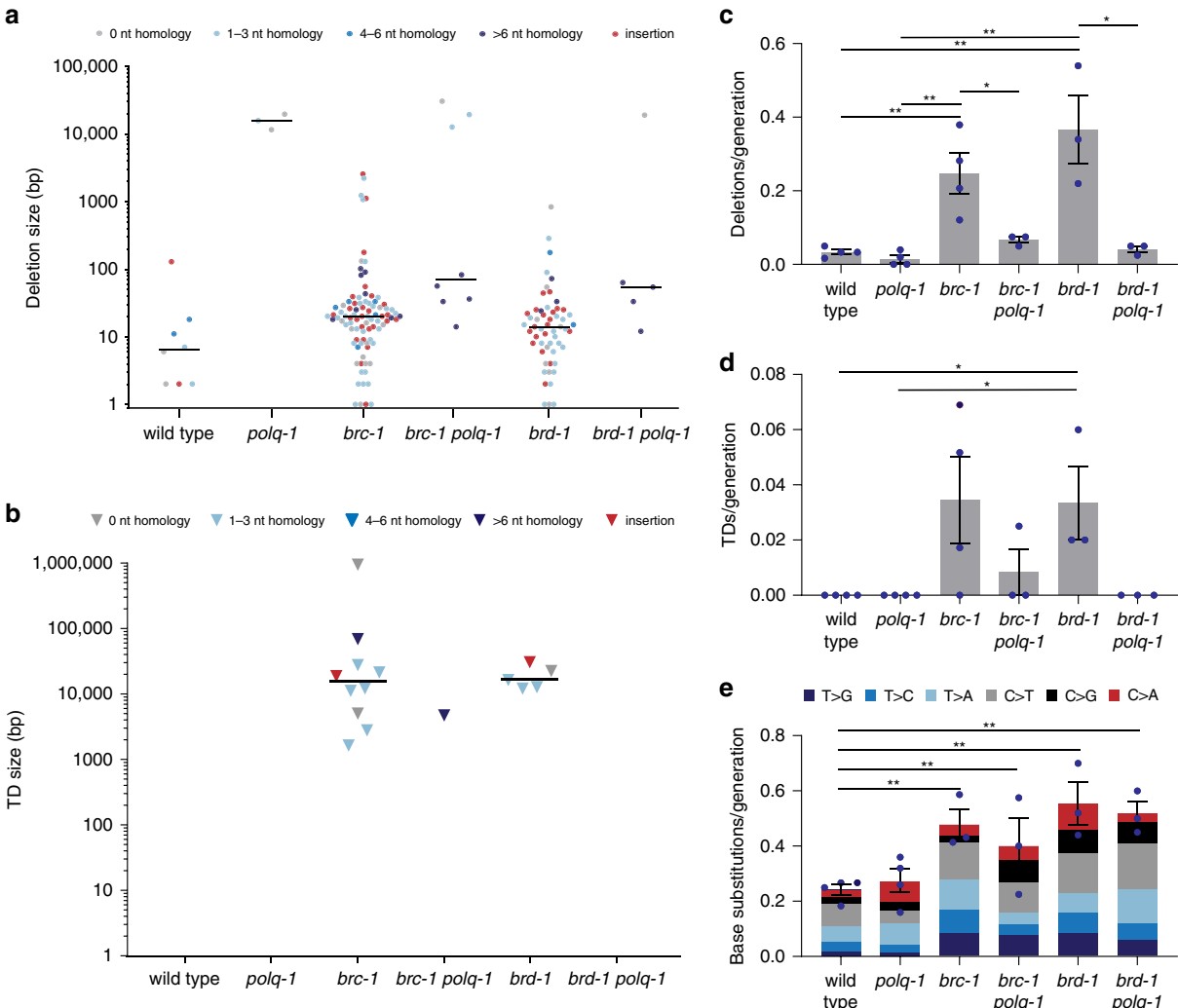

**Fig. 3 Knockout of polymerase theta alters deletion and TD rate and homology usage. a** Size and junction representation of deletions that were obtained in wild-type and mutant animals (*polq-1*: n = 4, *brc-1 polq-1* and *brd-1 polq-1*: n = 4). Deletions without homology are shown in grey, deletions with homology are marked in blue. Increasing homology size is depicted in increasing colour intensity. Deletions with insertions are marked in red. The median deletion sizes are indicated by horizontal lines. **b** Size and junction representation of tandem duplications (TDs) that were obtained in wild-type and mutant animals. TDs with homology at the junctions are marked in blue. Increasing homology size is depicted in increasing colour intensity. TDs with insertions are marked in red. The median TD sizes are indicated by horizontal lines. **c** Quantification of the average rate of deletions per generation in animals of different genotypes. The rate is defined as the number of deletions divided by the number of propagated generations per animal. The rate per strain is represented in blue dots. Two-tailed *t*-tests were used to determine statistical significance (*P < 0.05, **P < 0.01). **d** A quantification of the average rate of TDs per generation in animals of different genotypes. The rate per strain is represented in blue dots. The rate is defined as the number of TDs divided by the number of propagated generations per animal. Two-tailed *t*-tests were used to determine statistical significance (*P < 0.05, **P < 0.01). **e** Representation of base substitution type and the average rate of base substitutions per generation in animals of different genotypes. The different types of base substitutions are labelled with different colours. The rate is defined as the number of base substitutions divided by the number of propagated generations per animal. The rate per strain is represented in blue dots. Two-tailed *t*-tests were used to determine statistical significance (*P < 0.05, **P < 0.01). Error bars represent SEM. Wild type: n = 4 of 60 generations, *polq-1*: n = 4 of 50 generations, *brc-1*: n = 6 of 58 generations, *brd-1*: n = 3 of 50 generations, *brc-1 polq-1* and *brd-1 polq-1*: n = 3 of 40 generations.

upon invasion into the sister chromatid, to the other original DSB end. In that view, and in perfect agreement with previously proposed models[23,26], TMEJ acts to replace a late step in SDSA, with mutagenic consequences. The accumulation of both structural variations, i.e. deletions and TDs, in *brc-1*/*brd*-1 animals is most parsimoniously explained by inferring inadequate resection of break ends. The finding that knockdown of resection protein CtIP in mammalian cells leads to TDs to a similar extent as knockdown of BRCA1 and BARD1 supports this hypothesis[13,43]. Limited resection impairs the formation of a recombinogenic intermediate, yet allows for the detection and usage of micro-homologous sequences present at both ends of the break to guide

TMEJ. Such alternative repair would manifest as genomic deletions. In cases where resection has been restricted but sufficient to set up a RAD51 filament, strand invasion can take place, so does D-loop extension. However, upon D-loop disassembly the limited resection in BRCA1-deficient cells impairs the reannealing of the extended strand, as described by Chandramouly and colleagues[43]. In such a scenario, TMEJ can join the terminal sequence of the extended break end to that of the receiving break end to produce a TD (Fig. 5). The overrepresentation of micro-homology and the occasional presence of template insertions at the junction, as well as the near-absolute dependence on polymerase theta argues that TMEJ is acting on such SDSA intermediates in *brc-1*/*brd-1-*

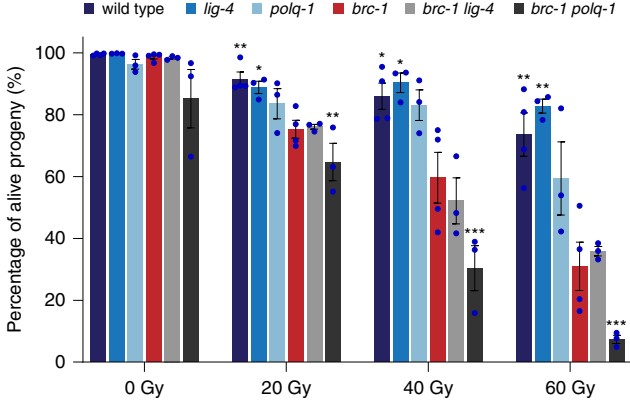

**Fig. 4 Worms lacking BRC-1 and POLQ-1 are hypersensitive to ionising irradiation (IR).** Wild type (dark blue), *lig-4* (blue), *polq-1* (light blue), *brc-1* (red), *brc-1 lig-4* (grey) and *brc-1 polq-1* (dark grey) L4 larvae were exposed to 0, 20, 40 or 60 Gy IR (*n* = 3, wild type and *brc-1: n* = 4). Three days after irradiation, the numbers of alive offspring versus unviable eggs were quantified. The average survival in each experiment is represented in blue dots. Statistical analysis shows significant differential loss of viability between *brc-1 polq-1* and *brc-1* animals (two-tailed *t*-tests vs. *brc-1* mutant *P < 0.05; **P < 0.01, ***P < 0.001). Error bars represent SEM.

deficient animals. It should be noted that this model also allows for (experimentally unnoticeable) error-free HR in case the ends are sufficiently processed to produce one recombinogenic DSB end and one SDSA-permitting DSB end.

One outcome that is predicted by this model is the absence of TD formation if HR is compromised such that RAD51 loading is completely lost, as the sister chromatid provides the template for the DNA that is duplicated. Unfortunately, mutations in *C. elegans* RAD51 or BRCA2 cannot be assayed as null mutants produce inviable gametes because of the essential role for these proteins during meiosis; BRC-1 is needed only for HR via the sister chromatid[44,45]. However, next-generation sequencing of tumour material have indeed revealed that TDs, in the same size-range as we have described here (~10 kb)[46], are very frequent in genomes of BRCA1-deficient tumour cells but not in tumours associated with BRCA2 loss[5,13,46,47].

Importantly and unexpectedly, our data also provide insight into the molecular mechanism of TMEJ as it further defines the precise requirement for polymerase theta's functions in joining two break ends together. Although the vast majority of BRC-1/ BRD-1 loss-associated small deletions and TDs are dependent on polymerase theta action, a few typical cases are not. These products all contain markedly larger stretches of homology (Fig. 3). The requirement for polymerase theta to join DSB ends can thus be bypassed by the presence of sequence homology surrounding the break that is just more than a few bases. This classifies TMEJ as a specific type of micro-homology-mediated end-joining (MMEJ): it requires less homologous nucleotides than other forms of MMEJ. Indeed, in systems that do not encode polymerase theta, MMEJ requires more extensive complementary sequences at the 3′ end of DSBs[48]. Our data thus reveal a quintessential function for polymerase theta in alternative end-joining: to extend a minute amount of sequence complimentary, that is sufficient to be used as a primer for polymerase theta action, to a stretch that is sufficient for other polymerases to act upon. Likely candidates for such a gap filling reaction are the replicative polymerases as they possess proofreading activity and thus prevent unnecessary mutations: in all manifestations of TMEJ we hardly find mutations in the sequences flanking the break[21]. Such a hand-over nicely fits the biochemical properties of replicative

polymerases which in vitro require a ~5–7 bp primer to start polymerisation without activating its proofreading activity[49].

Although our data provide insight into the processing of HR intermediates in *brc-1/brd-1* animals, their source is currently unknown. The size-range of the identified deletions is comparable to that of deletions that result from TMEJ repair of DSBs induced in the *C. elegans* germline through CRISPR/Cas9 action[31]. This size similarity could argue that the spontaneously occurring deletions in BRC-1/BRD-1 mutants are the consequence of near-blunt (or blunt) DSBs. These DSBs may result from SPO-11 activity during the meiotic stage of gametogenesis, also as these were found to persist in germ cells of *brc-1* mutants (Supplementary Fig. 2). However, it should be noted that the rate with which structural variations are induced—~1 per 4 animal generations—is significantly lower than the number of foci that can be seen in nuclei at late pachytene stages of meiotic profase (~3–4 per nucleus, Supplementary Fig. 2). Given the small size of the deletions, disruption of essential genes and consequent negative selection could only explain a very small part of this discrepancy. The discrepancy argues that not all of the foci are processed by TMEJ to form a deletion or a TD, and if not all, why any? What are alternative fates for these HR intermediates that become clearly apparent during gametogenesis? Certainly not corrupting embryogenesis as the degree of embryonic lethality in *brc-1* animals is negligible. Perhaps, intersister HR is not entirely abrogated in *brc-1/brd-1* null mutants. Our reporter data of DSB repair in somatic cells argue that BRC-1 is essential to repair the vast majority of DSBs in a context that reflects an SDSA mechanisms (Supplementary Fig. 1), but it does not exclude other forms of resolution of HR intermediates. An alternative explanation for the discrepancy between the numbers of persistent RAD51 foci in mitotic germ nuclei and the rate of mutagenesis is that the apoptotic programme acting prior to gamete maturation[50] may remove the majority of nuclei that have persistent DSBs. Indeed, apoptosis is elevated in the germline of *brc-1* animals[44]. It may also be that the mutagenesis we found does not have a meiotic origin, but instead results from spontaneous breaks or defective processing of stalled replication forks during mitotic growth, either of primordial germ cells or of those embryonic cells that contribute to the next animal generations. As to the origin of TDs, it was proposed that in addition to their formation via DSB-induced long tract gene conversion[43], a subset of BRCA1/BARD1-specific cases are produced via defective processing of stalled replication forks[13]. Both types require a final end-joining step to complete the TD. Willis and colleagues show an overrepresentation of micro-homology usage at the breakpoint of TDs induced by fork stalling[13], which is reminiscent of TMEJ.

A low rate of DSB induction when animals are grown in non-challenged conditions may explain the lack of overt developmental or growth problems of *brc-1* and *brd-1* mutants animals. TMEJ can repair HR substrates leading to mutations; in its absence compromised brood sizes and embryonic lethality starts to manifest. These phenotypes are logically exacerbated by inflicting additional DNA damage. The observation that TMEJ deficiency, while itself only marginally sensitising the worms to ionising radiation, strongly affects the survival of *brc-1* animals provides further support to the mutation data that TMEJ can compensate for loss of HR. This finding is in line with previous data in *Drosophila melanogaster*, where polymerase theta is required for the resistance to IR-induced damage in the absence of Rad51[23].

The remarkable similarity in mutation profiles in BRCA1-deficient tumour cells and BRCA1-deficient *C. elegans* suggests that the mechanisms leading to these mutations are evolutionarily conserved. A dependence on polymerase theta for HR-deficient cells was noticed earlier in mouse embryonic fibroblasts[51] and in

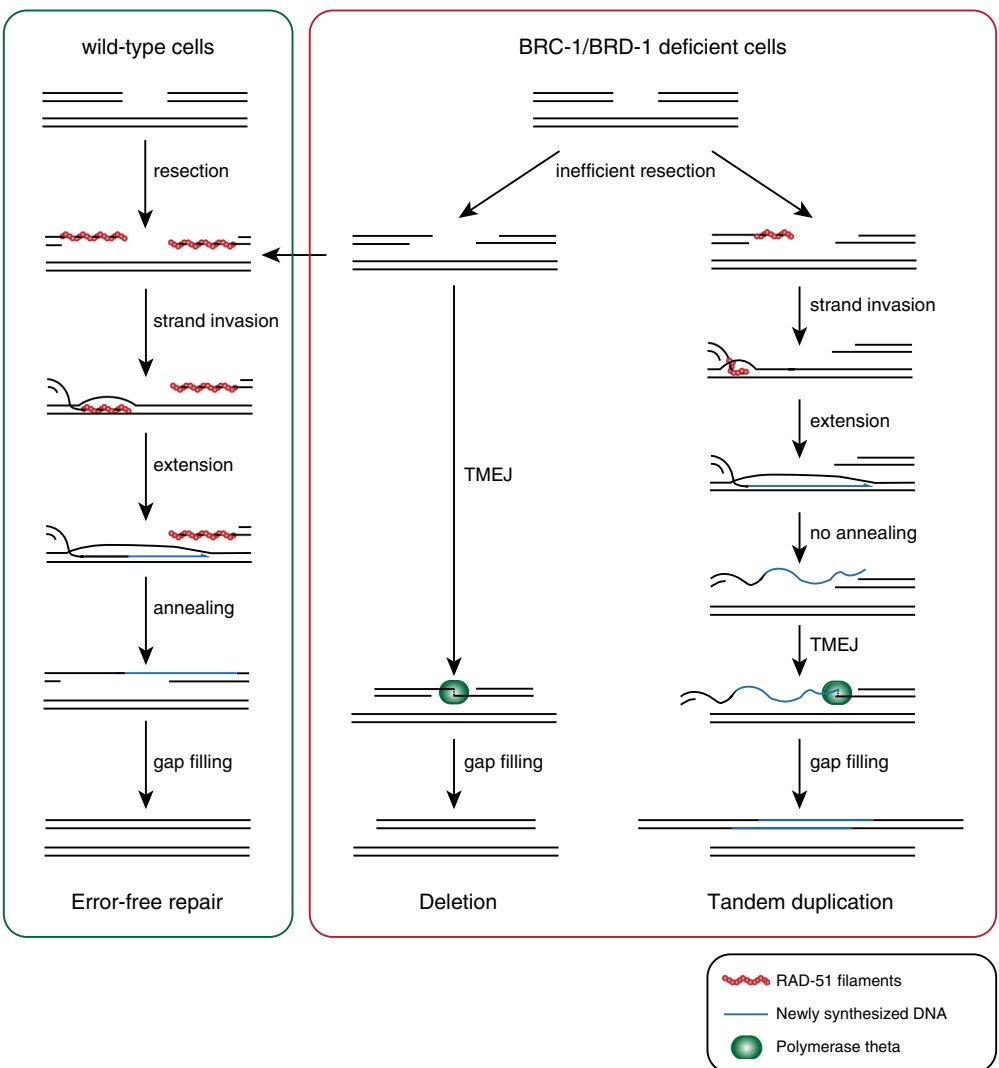

**Fig. 5 Model in BRC-1/BRD-1-deficient cells, inefficient resection leads to impaired strand invasion.** When strand invasion cannot occur, break ends are joined by polymerase theta. When strand invasion does occur, reannealing of the extended invaded strand can be hindered by insufficient resection of the receiving strand, again leading to direct end-joining by polymerase theta. When both break ends are sufficiently resected, error-free repair can take place, similar to repair in wild-type cells.

tumour cells: HR-deficient epithelial ovarian cancer cells depend on TMEJ[52], whereas similar effects were recently observed for cisplatin-resistant lung cancer cells (which had upregulated polymerase theta expression)[53]. Moreover, it was recently shown that tumours with *BRCA* mutations show a significantly higher frequency of templated insertions, which is a hallmark of polymerase theta action, than *BRCA* proficient tumours[54]. The dependency of HR-deficient cancer cells on TMEJ point to polymerase theta as an interesting therapeutic strategy for a specific set of tumours. We propose that these tumours can be identified using mutational footprint analysis as we have here demonstrated that the evolutionary conserved BRCAness signature is the product of TMEJ.

## Methods

***C. elegans* genetics**. Nematodes were cultured on standard NGM plates seeded with OP50 bacteria[55] at 20 °C. Bristol N2 was used as wild type. The following alleles were used in this study: *brd-1(dw1)*, *cku-80(ok861)*, *lig-4(ok716)* and *polq-1 (tm2026)*. Using CRISPR/Cas9 mutagenesis a novel *brc-1* allele (*lf249*, Supplemental Table 1) was generated (guide RNA sequence 5′-ACTGAGGATCACA-GAAACAG-3′) by injecting plasmids in N2 germlines using standard *C. elegans* microinjection procedures. The plasmids encode Cas9, guide RNA and mCherry marker, the latter to enable phenotypic selection of transgenic F1 progeny. The novel *brc-1* animals show increased levels of persistent RAD51 foci in their germlines (Supplementary Fig. 2), indicating a defect in the processing of meiotic breaks as expected in *brc-1* animals[44]. To obtain *brc-1 polq-1* and *brd-1 polq-1* double mutants, *polq-1(lf265)* and *polq-1(lf257)* were generated using the same guide RNA (5′-GCAGATTGATGTGTTGAATG-3′) in *brc-1* (*lf249*) and *brd-1* (*dw1*) mutants, respectively (Supplementary Table 1).

**Mutation accumulation assays**. Mutation accumulation lines were generated by cloning out ten F1 animals from one hermaphrodite. DNA from strains that were acquired via crossing was isolated at the start of each experiment (generation 0), to prevent heterozygous mutations present at the start of the experiment to be picked up as novel mutations. Each generation three nematodes were transferred to new plates. MA lines were maintained for 40–60 generations. Single animals from the last generation were transferred to new NGM plates. When sufficient offspring was present on these plates, nematodes were washed off with $H_2O$ and incubated for 2 h while shaking to remove bacteria from their intestines. Genomic DNA was isolated using a Blood and Tissue Culture Kit (Qiagen). DNA was sequenced on an Illumina HiSeq platform (2 × 150 bp paired-end reads). The average sequencing depth of the samples was 44.2 (Supplementary Table 2).

**Bioinformatic analysis**. Image analysis, base calling and error calibration were performed using standard Illumina software. Raw reads were mapped to the *C. elegans* reference genome (Wormbase release 235) by BWA[56] and SAMtools[57]. GATK[58] was used for SNV calling. Only unique homozygous SNVs across samples with GQ ≥ 40 and DP ≥ 8 were included. Pindel[59], GATK[58], Manta[60] and GRIDSS[61] were used for calling deletions and TDs. Variations were considered as

true events if they were covered by both forward and reverse reads and supported by at least five reads. In addition, reads supporting the reference should be <100. Events were only considered if they were uniquely present in one of the samples. All events were inspected by IGV[62] to ensure correctness of the call. Consistency in number of events between independent lines was verified (Supplementary Figs. 3 and 4).

**IR sensitivity assays**. L4 stage nematodes were exposed to ionising irradiation or mock-treated. Per experimental condition, three-seeded NGM plates containing three nematodes were prepared. The three irradiated nematodes were removed from the plate after a 48-h egg laying period. The number of hatched and unhatched progeny was quantified 24 h after removal.

**HDR-reporter assay**. Homology-driven repair was read out using a GFP reporter[30]. In brief, nematode populations carrying this reporter were synchronised by incubating the nematodes in a 3:2 mixture of hypochlorite (Acros Organics) to 4 M NaOH until worms were dissolved and only eggs remained. Eggs were washed with M9 buffer (22 mM $KH_2PO_4$, 42 mM $Na_2HPO_4$, 86 mM NaCl, 1 mM $MgSO_4$) to remove the bleach mixture. The residual eggs were allowed to hatch overnight in M9 buffer. After hatching, larvae were plated on seeded NGM plates and incubated at 34 °C for 2 h (heat shock). 24 hours after heat shock, nematodes were checked for IsceI-mCherry expression using a Leica DM6000 microscope. 72 h after heat shock, nematodes were scored for GFP-positive intestinal nuclei with a Leica DM6000 microscope ×10 objective.

**Germline immunostaining**. Germlines were dissected and isolated from young adults on a 18 × 18 mm coverslip, and subsequently fixed on a Superfrost Plus slide by freeze-cracking, methanol (−20 °C) freezing, 4% formaldehyde fix treatment and washing with PBST (1× PBS + 0.1% Tween-20). After blocking with PBST + 0.5% BSA, slides were incubated in a humid chamber overnight with 1:10,000 monoclonal anti-RAD51 antibody (Novus biologicals). Slides were subsequently washed twice with PBST, and then incubated with a 1000× diluted, secondary Alexa488-labelled goat-anti-rabbit antibody (Invitrogen) for 2 h. Slides were then washed thrice in PBST and subsequently DAPI-treated for 10 min. Slides were then washed for one hour in PBST and rinsed with 10 mM Tris pH 7.7 + 0.1% Tween-20 for 5 min. Finally, germlines were mounted with Vectashield. Stained germlines were visualised using a Zeiss Axio microscope coupled to an HXP 120V Illuminator. Z-stacks were obtained with a ×100 objective, using 0.5 μm distances.

**Reporting summary**. Further information on research design is available in the Nature Research Reporting Summary linked to this article.

## Data availability
Raw sequences have been made publicly available at NCBI SRA (accession code PRJNA599297). Data for N2 wild-type and *polq-1* animals were published previously and can be found at NCBI SRA (accession codes PRJNA260487 and PRJNA196232). A list of all events and raw data are available in Supplementary Data 1 and 2. No unreported custom computer code was used during this study. We used the *C. elegans* reference genome (Wormbase release 235) in this study. All data are available from the authors upon reasonable request.

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

## Acknowledgements

Some strains were provided by the CGC, which is funded by the NIH Office of Research Infrastructure Programs (P40 OD010440). We thank Ron Romeijn for technical assistance and Gabriele Patti for generating the *brc-1* and *brc-1 polq-1* mutants. J.K. was supported by ZonMW/NGI-Horizon and R.v.S. and M.T. are supported by NWO/ALW.

## Author contributions

J.K. and M.T. conceived and designed the study. J.K. and I.D. performed the experiments. R.v.S. performed the bioinformatics analysis. All authors interpreted the experimental data. J.K. and M.T. wrote the manuscript.

## Competing interests

The authors declare no competing interests.
