## [Peer Review File · Nature Communications]

Reviewers' comments:

Reviewer #1 (Remarks to the Author):

In this investigation, Kamp et al. report that *C. elegans* lacking the mammalian BRCA1 and BARD1 orthologs accumulate mutations similar to those seen in human *brca1*-deficient tumors. These mutations include relative small (<30 bp) deletions and large (1-1000kb) tandem duplications, both of which have microhomologous sequences at the breakpoints and/or small insertions. In addition, an increase in base substitutions is observed. They also show that polymerase theta mediated end joining, but not classical non-homologous end joining, is responsible for the majority of the deletions and tandem duplications.

While the role of POLQ in promoting deletions with accompanying microhomology is well-established, its involvement in the formation of tandem duplications has not been reported. This is the novel aspect of this paper. The authors propose that the tandem duplications could result from aborted homologous recombination, when one strand invades into a homologous template and DNA synthesis occurs, but the nascent strand is unable to anneal with the other broken end. They argue that this would be more likely in the absence of BRC-1 and BRD-1, due to a defect in resection.

My major concern is that while this model is attractive, the authors provide no direct support. They do cite other analyses showing that tandem duplications are common in human tumors with *brca1* mutations, but not with *brca2* mutations, but this is only correlative data. Because the main novelty of this study lies in the proposed mechanism of tandem duplication formation, some experimental support of their model is needed. The most obvious prediction, that a *brc-2* mutation would prevent their formation, cannot be tested due to the necessity of *brc-2* for meiotic recombination. Perhaps the authors can look in an *mre11 brc-1* double mutant (although these have reduced fertility)—the prediction would be that tandem duplications should also be more frequent due to a defect in resection. Alternatively, a mutation or condition that increases the amount of resection at double-strand breaks should decrease the frequency of tandem duplications. The inclusion of additional data to elucidate the conditions which allow TMEJ-mediated tandem duplication formation would significantly strengthen the manuscript.

Other points:

1. The Scully lab proposed an alternative model for tandem duplication formation involving replication restart-bypass in *brca1* mutant cells (Willis et al., 2017, PMID 29168504). While the current manuscript does reference this work, a more thorough discussion of this model is warranted.
2. The model for TMEJ repair following aborted HR is not novel. This was suggested at least two previous publications: Wyatt et al., 2016 (PMID 27453047) and Chan et al., 2010 (PMID 20617203). These papers should be cited when the model is proposed.
3. A synergism between HR and TMEJ in the repair of IR-induced damage was previously reported in Chan et al., 2010. In this instance, mutation of both RAD51 and POLQ resulted in hypersensitivity to IR, similar to that reported in the current study. This should be cited.
4. Lines 98-99: More detail about the type of somatic cells and how HDR was monitored should be provided here, so the readers don't have to rely only on Figure S1.
5. Line 103: Are the accompanying insertions templated from flanking sequences, as has been reported for other TMEJ junctions?
6. Line 110: I seem to recall that earlier papers from this group showed that deletions that occur at G4 sequences almost always involved 1 nt of microhomology. Is the proportion different in *brc-1* mutants and if so, why might this be?

7. Lines 174-175: this statement is oversimplified, as it is true only for relatively small deletions (Fig. 3).

8. Lines 189-198: This paragraph is confusing as written. Are the authors arguing that TMEJ is more common in *C. elegans* than in mammals? The phrase "provides an unlikely explanation" makes this unclear.

9. Line 263: The authors claim that the reliance on short microhomologies sets TMEJ apart from MMEJ, but there are no consistent guidelines for MMEJ in the literature-- particularly as it is defined differently in yeast vs. other organisms, as the next sentence points out. MMEJ is often used as a general term for the use of any amount of microhomology during break repair. Therefore, it would be better to use different terminology here.

Reviewer #2 (Remarks to the Author):

The work of Kamp et al. proposes a key role for the DNA polymerase theta in generating mutation patterns that are known to be characteristic of homologous recombination (HR)-deficient human cells, using a worm germline mutagenesis model. These are important findings because they shed light on the mechanisms underlying mutagenesis under HR deficiency. An implication of these results is that they further suggest relevance of therapeutically targeting DNA polymerase theta in those HR-deficient tumors that exhibit certain mutation patterns. Overall the methodology appears quite solid and the presentation balanced. I do have a few queries that need to be addressed.

1. The conclusions have some potential to be affected by technical issues stemming from mutation calling from short reads, which is error-prone in case of calling short indels and moreso in case of larger structural variants.

- only a single tool to call structural variants is used (Pindel); more modern tools (e.g. Manta, Gridss etc) may give more accurate results, e.g. in terms of placing the breakpoint, or in terms of false negative rates; note that the latter cannot be prevented by manual inspection of calls which they have done. Running an additional caller would help ensure the results are not skewed towards biases of a single algorithm. In particular different tools may have different biases with respect to lengths/types of variants they can detect, which can be relevant for the statements such as line 117 "The rate of TDs in *brc-1* and *brd-1* mutants is approximately tenfold lower than the rate of deletions in these mutants."

- would the conclusions be affected if limiting to variant calls with higher confidence scores and/or in less difficult to align (less repetitive) genome regions?

- please state average and/or median depth of sequencing in the methods.

2. Data presentation issues.

- there is much overplotting of the dots in 1A/1B/2A/2B/3A/3B which makes it hard to judge the density of points of different colors (microhomology lengths), affecting interpretation of results. I suggest to sort the points (x axis) by # nucleotides of microhomology, within each genotype.

- line 142 "we observed a small increase in deletion mutagenesis in *cku-80* and *lig-4*" and line 146 "base substitution rate is mildly increased in *cku-80* and *lig-4* mutants as compared to wildtype". I would not say this is a small nor mild increase (≥ 2 fold in Figs 2C/2E). In 2E, the SNV rate increase from NHEJ KOs is bigger than for *brc-1* KO. Consider rephrasing.

- please draw two-sided error bars on bar charts

- mutation rates flanking the breaks do not appear to be elevated, arguing against common use of an error-prone polymerase to extend past pol theta activity. Pol theta itself does not appear to contribute many SNVs according to Fig 3e. Thus currently the cause of higher SNV mutation rates in *brc-1*/*brd-1* backgrounds is unclear. Would looking at the trinucleotide mutation spectrum, and examining correlations to known mutation signatures from human tumors, be helpful in elucidating this?

3. Multiple lines of animals k.o. for various genes were propagated, and mutation data pooled together for analysis. Are the results overall consistent between the independent lines? Potentially some of the observations might be driven by an acquired mutator in one of the lines (e.g. the increased SNV rate in NHEJ deficient lines).

4. Regarding negative selection. Lines 179-181 "Deletions of that size will affect multiple genes in a gene-dense organism such as *C. elegans* and their presence are likely counterselected for in animal propagation experiments". It is well possible that many of the smaller deletions are also under selection. Please discuss and/or address with analyses if this has the potential to bias results (e.g. there might be confounding between chromosomal locations of fitness genes and the chromosomal location bias of activity of the various HR failure-related mutational processes).

5. A 1 nt microhomology has a rather high probability of occurring at random (depending on the local nucleotide composition). On the one hand, this affects interpretation of the data shown in plots, where 1 nt microhomologies, often arising at random, are lumped together with 2 and 3 nt microhomologies, which are more likely to stem from real signal. On the other hand, this can affect interpretation of statements such as line 110: "These deletions are also characterized by an overrepresentation of micro-homology: 58% of deletions had at least one nucleotide that could be mapped to either junction". A baseline estimate of microhomology should be provided, if relying on 1 nt matches to call MHs.

Minor.

- typos: "undistinguishable", "we have here describe (~10 kb)". Line 169 might need to refer to Fig 3A.

- line 174: "Our data demonstrate that all deletions with ≤ 6 nt of homology are the result of polymerase theta on DSBs". This is correct, however if I understand correctly this 6nt threshold might be better placed at a higher number of nucleotides, depending on the lengths of the microhomology segments in polq-1 mutants in Fig 3A, which are not evident from the figure.

- line 107: "We found no significant difference in deletion rate between *brc-1* and *brd-1* ($p=0.236$), arguing against a BRC-1-independent role for BRD-1 in deletion prevention, and vice versa". I would not say this is necessarily very strong evidence, in the absence of experiments on the *brc-1 brd-1* double mutant, which they did not perform.

Response to the reviewers

First of all, we would like to thank both reviewers for their constructive comments and suggestions, which helped us to improve our manuscript.

Reviewer #1 (Remarks to the Author):

In this investigation, Kamp et al. report that C. elegans lacking the mammalian BRCA1 and BARD1 orthologs accumulate mutations similar to those seen in human brca1-deficient tumors. These mutations include relative small (<30 bp) deletions and large (1-1000kb) tandem duplications, both of which have microhomologous sequences at the breakpoints and/or small insertions. In addition, an increase in base substitutions is observed. They also show that polymerase theta mediated end joining, but not classical non-homologous end joining, is responsible for the majority of the deletions and tandem duplications.

While the role of POLQ in promoting deletions with accompanying microhomology is well-established, its involvement in the formation of tandem duplications has not been reported. This is the novel aspect of this paper.

While we certainly agree with the reviewer that the role of polymerase theta in promoting deletions with microhomology is well-established, it was not shown previously that polymerase theta-mediated end-joining (TMEJ) is responsible for deletions that specifically arise in absence of BRCA1. And while it was previously proposed that TMEJ could repair aborted homologous recombination (HR) products^{1,2}, models drawn in very recent papers in top tier journals depicting repair in a BRCA1-deficient context show non-homologous end-joining (NHEJ), not TMEJ, as the repair mechanism³⁻⁶.

Based on the published literature, one could a priori speculate that both NHEJ and TMEJ repair double strand breaks in BRCA1 deficient cells. The predominance of microhomology is a very strong indication but yeast cells employ MMEJ without encoding polymerase theta. Here, we are the first to provide evidence that polymerase theta, and no other EJ activity, is responsible for the vast majority of structural variations in a BRCA1-deficient context, and we feel this to be the major novel finding of our paper, hence it is this aspect that was (and still is) reflected in the title and abstract, where we do not speculate or make claims on the mechanism of tandem duplication formation.

The authors propose that the tandem duplications could result from aborted homologous recombination, when one strand invades into a homologous template and DNA synthesis occurs, but the nascent strand is unable to anneal with the other broken end. They argue that this would be more likely in the absence of BRC-1 and BRD-1, due to a defect in resection.

My major concern is that while this model is attractive, the authors provide no direct support. They do cite other analyses showing that tandem duplications are common in human tumors with brca1 mutations, but not with brca2 mutations, but this is only correlative data. Because the main novelty of this study lies in the proposed mechanism of tandem duplication formation, some experimental support of their model is needed. The most obvious prediction, that a brc-2 mutation would prevent their formation, cannot be tested due to the necessity of brc-2 for meiotic recombination. Perhaps the authors can look in an mre11 brc-1 double mutant (although these have reduced fertility)—the prediction would be that tandem duplications should also be more frequent due to a defect in resection. Alternatively, a mutation or condition that increases the amount of resection at double-strand breaks should decrease the frequency of tandem duplications. The inclusion of additional data

to elucidate the conditions which allow TMEJ-mediated tandem duplication formation would significantly strengthen the manuscript.

While we always strive to improve our manuscripts by including experiments suggested by reviewers, there are a number of remarks to be made concerning this.

Initial steps of TD formation were previously described in papers from the Scully lab^{7,8}, where the authors demonstrate that defects in resection can lead to TD formation. However, the final end-joining step was not elucidated. Our novel contribution to the TD formation model is thus not in the initial steps (involving resection), but in the last steps: joining an extended end to an opposing end, also suggesting a DSB as an intermediate. How TDs are formed, potentially as intermediates of a disturbed HR reaction, and whether one can titrate the frequency of TDs by regulating the amount of resection is an exciting idea for further investigation, but we feel that addressing this biology goes beyond the scope of the current manuscript.

But importantly, the suggested experiment is also not possible: *mre-11* deficient nematodes cannot reproduce to an extent that we can perform mutation accumulation experiment: 96-100% of the brood of homozygous mutants is not viable^{9,10}, making it impossible to maintain populations for generations (even for DNA isolation). As an alternative, we tried to sustain populations of *com-1* (CtiP) *ku-80* double mutants – *com-1* knockout are also inviable but lethality can be partly rescued by inactivating NHEJ – but also these genetic backgrounds do not tolerate (multigenerational) propagation.

As a final remark, our expectation of impairing MRE-11 is also different from the reviewer's prediction. The MRN complex is likely vital to initiate resection. In nematodes, it is also essential to create meiotic DSBs⁹, however a hypomorphic allele facilitates DSB formation but completely fails in resection¹⁰. We thus do not expect to observe more frequent duplications in *brc-1 mre-11* mutants (if they were to be viable), but instead no duplications, as there needs to be at least some resection to allow RAD-51 filament formation to set-up a D-loop. The HR defects in BRCA1, perhaps because of the multi-functionality of the protein, is likely unique and cannot easily, if at all, be mimicked by another (yet to define) genetic context. The consequences of loss of BRCA1 and how BRCA1 acts in DNA break repair has been the subject of intense research over the last 30 years and is still unanswered.

We would like to stress here that we have put forward a model in the discussion section to reconcile our data with all currently available data– we do not experimentally address this issue at all in our manuscript, and we do not mention anything related to this in the abstract and title or claim novelty. We have now amended the text on a number of occasions, helped by the suggestions of this reviewer, to make this clearer.

Other points:

*1. The Scully lab proposed an alternative model for tandem duplication formation involving replication restart-bypass in *brca1* mutant cells (Willis et al., 2017, PMID 29168504). While the current manuscript does reference this work, a more thorough discussion of this model is warranted.*

Indeed, in the paper by Willis et al. the Scully lab proposes an alternative origin of TDs in addition to their previous model in which they describe the formation of TDs by long-tract gene conversion⁷. This work is now included in our manuscript (lines 316-320). Furthermore, in lines 250-251 we now mention the Scully lab's data on the relation between resection impairment and TD formation^{7,8}.

2. *The model for TMEJ repair following aborted HR is not novel. This was suggested at least two previous publications: Wyatt et al., 2016 (PMID 27453047) and Chan et al., 2010 (PMID 20617203). These papers should be cited when the model is proposed.*

We corrected the manuscript to include these references (line 247). Moreover, after submission of our manuscript another paper from the Ramsden lab proposing TMEJ following aborted HR was published¹¹. We also included a reference to this work (lines 335-337).

3. *A synergism between HR and TMEJ in the repair of IR-induced damage was previously reported in Chan et al., 2010. In this instance, mutation of both RAD51 and POLQ resulted in hypersensitivity to IR, similar to that reported in the current study. This should be cited.*

We apologise for having forgotten to include this very important observation that fuelled the interest in polymerase theta. We now cited this paper when we discuss our IR data (line 327-329).

4. *Lines 98-99: More detail about the type of somatic cells and how HDR was monitored should be provided here, so the readers don't have to rely only on Figure S1.*

We now included additional information (lines 100-101).

5. *Line 103: Are the accompanying insertions templated from flanking sequences, as has been reported for other TMEJ junctions?*

Of all delins (46) found in *brc-1* and *brd-1* deficient nematodes, 33 of the insertions were at least 5 nucleotides long, allowing their origin to be determined with sufficient reliability. Of these insertions, 24 (73%) were verified to be templated insertions. This is now mentioned in lines 117-122 in the manuscript. Since templated insertions are known to be specific for TMEJ^{11,12}, this provides additional evidence for the essential role of polymerase theta in generating these delins.

6. *Line 110: I seem to recall that earlier papers from this group showed that deletions that occur at G4 sequences almost always involved 1 nt of microhomology. Is the proportion different in *brc-1* mutants and if so, why might this be?*

Correct: analyses of thousands of TMEJ scars that are induced in the *C. elegans* genome by a diverse range of replication-blocking DNA lesions or secondary structures show that ~80 % have at least 1 nucleotide of microhomology¹³. This is comparable to the percentage of microhomology-mediated deletions found here: 79% of the deletions in *brc-1* and 78% in *brd-1* show 1 nucleotide of homology. This is now mentioned in lines 112-115 in the manuscript.

7. *Lines 174-175: this statement is oversimplified, as it is true only for relatively small deletions (Fig. 3).*

We stand corrected and now amended this sentence (line 185).

8. *Lines 189-198: This paragraph is confusing as written. Are the authors arguing that TMEJ is more common in *C. elegans* than in mammals? The phrase "provides an unlikely explanation" makes this unclear.*

The paragraph was indeed rather unclear; we have now rephrased the text (lines 200-204).

9. *Line 263: The authors claim that the reliance on short microhomologies sets TMEJ apart from MMEJ, but there are no consistent guidelines for MMEJ in the literature-- particularly as it is defined differently in yeast vs. other organisms, as the next sentence points out. MMEJ is often used as a general term for the use of any amount of microhomology during break repair. Therefore, it would be better to use different terminology here.*

Also here we agree with the reviewer, in fact we ourselves strive towards precise descriptions and terminology that exclude ambiguity as much as possible, which made us coin the term TMEJ in the first place, when we found in worms that 20% of TMEJ cases not having any micro-homology (and 80% only 1 nucleotide being statistically supported). Qualifying such cases as MMEJ hence seems illogical. We now rephrased this sentence (lines 280-281).

Reviewer #2 (Remarks to the Author):

The work of Kamp et al. proposes a key role for the DNA polymerase theta in generating mutation patterns that are known to be characteristic of homologous recombination (HR)-deficient human cells, using a worm germline mutagenesis model. These are important findings because they shed light on the mechanisms underlying mutagenesis under HR deficiency. An implication of these results is that they further suggest relevance of therapeutically targeting DNA polymerase theta in those HR-deficient tumors that exhibit certain mutation patterns. Overall the methodology appears quite solid and the presentation balanced. I do have a few queries that need to be addressed.

1. The conclusions have some potential to be affected by technical issues stemming from mutation calling from short reads, which is error-prone in case of calling short indels and more so in case of larger structural variants.

- only a single tool to call structural variants is used (Pindel); more modern tools (e.g. Manta, Gridss etc) may give more accurate results, e.g. in terms of placing the breakpoint, or in terms of false negative rates; note that the latter cannot be prevented by manual inspection of calls which they have done. Running an additional caller would help ensure the results are not skewed towards biases of a single algorithm. In particular different tools may have different biases with respect to lengths/types of variants they can detect, which can be relevant for the statements such as line 117 "The rate of TDs in brc-1 and brd-1 mutants is approximately tenfold lower than the rate of deletions in these mutants."

We thank the reviewer for this suggestion. We have re-run the analyses with Manta, GRIDSS and GATK, besides Pindel, and again checked the called variants using IGV. Using these programs, we identified twelve additional deletions and one delins that were not identified by Pindel. The sizes of these events ranged from 2 to 18767 nt. No new tandem duplications were identified. We conclude that our results were not biased by the use of Pindel, and our conclusions remain valid, yet we updated our figures to include the newly identified variants.

- would the conclusions be affected if limiting to variant calls with higher confidence scores and/or in less difficult to align (less repetitive) genome regions?

Because we study germline variants (and not somatic events), which become homozygous upon prolonged culturing, we already have high confidence scores. We also check for a drop or rise of coverage, for indels and TDs respectively. We exclude highly repetitive regions.

- please state average and/or median depth of sequencing in the methods.

The average coverage is 44.2. We now stated this in the method section (line 355) and added the coverage per sample in the supplemental data.

2. Data presentation issues.

- there is much overplotting of the dots in 1A/1B/2A/2B/3A/3B which makes it hard to judge the density of points of different colors (microhomology lengths), affecting interpretation of results. I suggest to sort the points (x axis) by # nucleotides of microhomology, within each genotype.

We updated our figures to reduce overplotting by making dot-sizes smaller. We have also added the suggested plots (homology on x-axis) in the supplemental data, as these can bring additional clarity on subtleties, perhaps most importantly the lack of cases with not a single base of micro-homology. We nevertheless prefer the original format to the suggested visualization for the main text as these highlight the profound difference between the genotypes more clearly (to the average reader – we asked our colleagues for feedback on this issue).

- line 142 “we observed a small increase in deletion mutagenesis in *cku-80* and *lig-4*” and line 146 “base substitution rate is mildly increased in *cku-80* and *lig-4* mutants as compared to wildtype”. I would not say this is a small nor mild increase (≥ 2 fold in Figs 2C/2E). In 2E, the SNV rate increase from NHEJ Kos is bigger than for *brc-1* KO. Consider rephrasing.

We rephrased these sentences (lines 152 and 156).

- please draw two-sided error bars on bar charts

This is now corrected.

- mutation rates flanking the breaks do not appear to be elevated, arguing against common use of an error-prone polymerase to extend past *pol theta* activity. *Pol theta* itself does not appear to contribute many SNVs according to Fig 3e. Thus currently the cause of higher SNV mutation rates in *brc-1/brd-1* backgrounds is unclear. Would looking at the trinucleotide mutation spectrum, and examining correlations to known mutation signatures from human tumors, be helpful in elucidating this?

We indeed also considered performing this analysis previously. However, the number of SNVs does not provide enough power to perform this analysis.

3. Multiple lines of animals k.o. for various genes were propagated, and mutation data pooled together for analysis. Are the results overall consistent between the independent lines? Potentially some of the observations might be driven by an acquired mutator in one of the lines (e.g. the increased SNV rate in NHEJ deficient lines).

We previously checked the independent samples for potential differences, but found none standing above what can be explained by a stochastic nature/rate of mutagenesis. This is also the conclusion after incorporation of the new variant calling tools (see graph below). Importantly, sample XF1319_BEG58 consists of a pooled sample from three lines (B, E and G), therefore the number of detected structural variations is higher than for the other samples (this was done to test whether we can reliably identify structural variations in pools to eventually save costs of WGS).

Also for SNVs, the samples of each genotype are comparable:

4. Regarding negative selection. Lines 179-181 "Deletions of that size will affect multiple genes in a gene-dense organism such as *C. elegans* and their presence are likely counterselected for in animal propagation experiments". It is well possible that many of the smaller deletions are also under selection. Please discuss and/or address with analyses if this has the potential to bias results (e.g. there might be confounding between chromosomal locations of fitness genes and the chromosomal location bias of activity of the various HR failure-related mutational processes).

This is something we cannot rule out, but suspect to only marginally effect our results: only 10% of *C. elegans* genes are essential for viability. It is also not obvious from monitoring brood-sizes and fertility, which for *brc-1* and *brd-1* mutants are comparable to wild type. We nevertheless now included this notion in the discussion section (line 301-302).

5. A 1 nt microhomology has a rather high probability of occurring at random (depending on the local nucleotide composition). On the one hand, this affects interpretation of the data shown in plots, where 1 nt microhomologies, often arising at random, are lumped together with 2 and 3 nt microhomologies, which are more likely to stem from real signal. On the other hand, this can affect interpretation of statements such as line 110: "These deletions are also characterized by an overrepresentation of micro-homology: 58% of deletions had at least one nucleotide that could be mapped to either junction". A baseline estimate of microhomology should be provided, if relying on 1 nt matches to call MHs.

We agree with the reviewer. Previously, we calculated and empirically determined that the expected percentage of deletions with one or more nucleotides of homology is 47% in the *C. elegans* genome¹³. We nevertheless found the percentage of the simple deletions with ≥ 1 nt homology to be 79% (deletions called with the new variant calling tools are included), which is in perfect agreement with our previously published data on thousands TMEJ scars induced by DNA damage. We added this in the text (lines 112-116).

Minor.

- typos: "undistinguishable", "we have here describe (~10 kb)". Line 169 might need to refer to Fig 3A.

We thank the reviewer for pointing out these typos, we corrected them.

- line 174: "Our data demonstrate that all deletions with ≤ 6 nt of homology are the result of polymerase theta on DSBs". This is correct, however if I understand correctly this 6nt threshold might be better placed at a higher number of nucleotides, depending on the lengths of the microhomology segments in *polq-1* mutants in Fig 3A, which are not evident from the figure.

The small deletions that appear in the *brc-1 polq-1* and *brd-1 polq-1* have 7 (2x), 8 (3x), 9, 11, 14 and 24 nucleotides of micro-homology at the junction, while one deletion with 6 nt of micro-homology was found in *brc-1*. We therefore set the cutoff at 6. We understand this was not clear in the previous version of our manuscript. We therefore now include a supplemental figure (see below).

- line 107: "We found no significant difference in deletion rate between *brc-1* and *brd-1* ($p=0.236$), arguing against a BRC-1-independent role for BRD-1 in deletion prevention, and vice versa". I would

not say this is necessarily very strong evidence, in the absence of experiments on the *brc-1 brd-1* double mutant, which they did not perform.

We agree that a double mutant would strengthen this claim. However, we feel this statement is still technically correct. We altered this sentence to tone down our claim (lines 109-110).

- 1 Wyatt, D. W. *et al.* Essential Roles for Polymerase theta-Mediated End Joining in the Repair of Chromosome Breaks. *Molecular cell* **63**, 662-673 (2016).
- 2 Chan, S. H., Yu, A. M. & McVey, M. Dual roles for DNA polymerase theta in alternative end-joining repair of double-strand breaks in *Drosophila*. *PLoS Genet* **6**, e1001005, doi:10.1371/journal.pgen.1001005 (2010).
- 3 Noordermeer, S. M. *et al.* The shieldin complex mediates 53BP1-dependent DNA repair. *Nature* **560**, 117-121 (2018).
- 4 Ghezraoui, H. *et al.* 53BP1 cooperation with the REV7-shieldin complex underpins DNA structure-specific NHEJ. *Nature* **560**, 122-127 (2018).
- 5 Mirman, Z. & de Lange, T. 53BP1: a DSB escort. *Genes Dev* **34**, 7-23 (2020).
- 6 Setiাপutra, D. & Durocher, D. Shieldin - the protector of DNA ends. *EMBO reports* **20** (2019).
- 7 Chandramouly, G. *et al.* BRCA1 and CtIP suppress long-tract gene conversion between sister chromatids. *Nat Commun* **4**, 2404, doi:10.1038/ncomms3404 (2013).
- 8 Willis, N. A. *et al.* Mechanism of tandem duplication formation in BRCA1-mutant cells. *Nature* **551**, 590-595, doi:10.1038/nature24477 (2017).
- 9 Chin, G. M. & Villeneuve, A. M. *C. elegans mre-11* is required for meiotic recombination and DNA repair but is dispensable for the meiotic G(2) DNA damage checkpoint. *Genes Dev* **15**, 522-534 (2001).
- 10 Yin, Y. & Smolikove, S. Impaired resection of meiotic double-strand breaks channels repair to nonhomologous end joining in *Caenorhabditis elegans*. *Mol Cell Biol* **33**, 2732-2747 (2013).
- 11 Carvajal-Garcia, J. *et al.* Mechanistic basis for microhomology identification and genome scarring by polymerase theta. *Proceedings of the National Academy of Sciences of the United States of America*, doi:10.1073/pnas.1921791117 (2020).
- 12 Schimmel, J., van Schendel, R., den Dunnen, J. T. & Tijsterman, M. Templated Insertions: A Smoking Gun for Polymerase Theta-Mediated End Joining. *Trends Genet* **35**, 632-644, doi:10.1016/j.tig.2019.06.001 (2019).
- 13 Koole, W. *et al.* A Polymerase Theta-dependent repair pathway suppresses extensive genomic instability at endogenous G4 DNA sites. *Nat Commun* **5**, 3216, doi:10.1038/ncomms4216 (2014).

REVIEWERS' COMMENTS:

Reviewer #1 (Remarks to the Author):

In their revision, Kamp et al. have addressed most of my concerns. I agree that their study nicely shows that TMEJ is responsible for deletions that specifically arise in the absence of BRCA1. However, I am still not convinced by their claim of novelty. A study with human cells showed that reduction of BRCA1 resulted in a deletion-prone signature that was dependent on CtIP, MRE11, POLQ, and PARP (Ahrabi et al., NAR 2016, PMID: 27131361). This paper is not cited in the revision. Nonetheless, I think that the value added here is that Kemp and colleagues observe a similar phenomenon in a whole organism context. In addition, they have studied it in the context of spontaneous mutagenesis, which is a novel angle.

The revisions to the text and figures have improved the readability and clarity of the manuscript. Specifically, the rewrite of the TMEJ vs. MMEJ section in the discussion is much improved. ("This classifies TMEJ as a specific type of microhomology-mediated end joining (MMEJ): it requires less homologous nucleotides than other forms of MMEJ"). I appreciate the authors' perspective on this.

One final suggestion: I still don't understand the logic in the introduction to the ionizing radiation section, or why the authors feel it necessary to explain why *C. elegans* brca1 mutants are viable but human cells lacking BRCA1 die (lines 200-204). There are many examples of DNA repair genes that are essential in vertebrates but not in invertebrates. The authors might consider simply pointing out the brca1 phenotypic difference between mammals and worms and then state that they wanted to further investigate a potential synthetic lethal interaction between HR and TMEJ, as has been shown in human cells.

Reviewer #2 (Remarks to the Author):

The authors have addressed my remarks satisfactorily. I have no further queries. I can recommend this manuscript for publication.

REVIEWERS' COMMENTS:

Reviewer #1 (Remarks to the Author):

In their revision, Kamp et al. have addressed most of my concerns. I agree that their study nicely shows that TMEJ is responsible for deletions that specifically arise in the absence of BRCA1. However, I am still not convinced by their claim of novelty. A study with human cells showed that reduction of BRCA1 resulted in a deletion-prone signature that was dependent on CtIP, MRE11, POLQ, and PARP (Ahrabi et al., NAR 2016, PMID: 27131361). This paper is not cited in the revision.

The cited paper concerns repair of induced DSBs, not spontaneous mutation accumulation leading to BRCAness, but perhaps more importantly (and remarkably, given the wording in the abstract of this paper), BRCA1 and POLQ were not co-depleted in this study. Proof for a causal link between repair outcomes in BRCA1-depleted cells and POLQ activity was thus not provided. Nevertheless, we now cite the paper in the introduction (line 69).

Nonetheless, I think that the value added here is that Kemp and colleagues observe a similar phenomenon in a whole organism context. In addition, they have studied it in the context of spontaneous mutagenesis, which is a novel angle.

The revisions to the text and figures have improved the readability and clarity of the manuscript. Specifically, the rewrite of the TMEJ vs. MMEJ section in the discussion is much improved. (“This classifies TMEJ as a specific type of microhomology-mediated end joining (MMEJ): it requires less homologous nucleotides than other forms of MMEJ”). I appreciate the authors’ perspective on this.

One final suggestion: I still don’t understand the logic in the introduction to the ionizing radiation section, or why the authors feel it necessary to explain why *C. elegans* brca1 mutants are viable but human cells lacking BRCA1 die (lines 200-204). There are many examples of DNA repair genes that are essential in vertebrates but not in invertebrates. The authors might consider simply pointing out the brca1 phenotypic difference between mammals and worms and then state that they wanted to further investigate a potential synthetic lethal interaction between HR and TMEJ, as has been shown in human cells.

We feel it is important to address this issue, because it tells the audience more about the validity of using *C.elegans* for addressing the research question.

Reviewer #2 (Remarks to the Author):

The authors have addressed my remarks satisfactorily. I have no further queries. I can recommend this manuscript for publication.

We thank both reviewers for reviewing our manuscript.